# Visual-Inertial Odometry Using High Flying Altitude Drone Datasets †

Anand George *, Niko Koivumäki, Teemu Hakala, Juha Suomalainen and Eija Honkavaara

Department of Remote Sensing and Photogrammetry, Finnish Geospatial Research Institute, 02150 Espoo, Finland
* Correspondence: anand.george@maanmittauslaitos.fi
† This article is based on the Master's thesis of the first author Anand George, available at http://urn.fi/URN:NBN:fi:aalto-2021121910926 (accessed on 1 February 2022).

**Abstract:** Positioning of unoccupied aerial systems (UAS, drones) is predominantly based on Global Navigation Satellite Systems (GNSS). Due to potential signal disruptions, redundant positioning systems are needed for reliable operation. The objective of this study was to implement and assess a redundant positioning system for high flying altitude drone operation based on visual-inertial odometry (VIO). A new sensor suite with stereo cameras and an inertial measurement unit (IMU) was developed, and a state-of-the-art VIO algorithm, VINS-Fusion, was used for localisation. Empirical testing of the system was carried out at flying altitudes of 40–100 m, which cover the common flight altitude range of outdoor drone operations. The performance of various implementations was studied, including stereo-visual-odometry (stereo-VO), monocular-visual-inertial-odometry (mono-VIO) and stereo-visual-inertial-odometry (stereo-VIO). The stereo-VIO provided the best results; the flight altitude of 40–60 m was the most optimal for the stereo baseline of 30 cm. The best positioning accuracy was 2.186 m for a 800 m-long trajectory. The performance of the stereo-VO degraded with the increasing flight altitude due to the degrading base-to-height ratio. The mono-VIO provided acceptable results, although it did not reach the performance level of the stereo-VIO. This work presented new hardware and research results on localisation algorithms for high flying altitude drones that are of great importance since the use of autonomous drones and beyond visual line-of-sight flying are increasing and will require redundant positioning solutions that compensate for potential disruptions in GNSS positioning. The data collected in this study are published for analysis and further studies.

**Data Set:** https://doi.org/10.23729/f4c5d2d0-eddb-40a1-89ce-601c252dab35.

**Keywords:** outdoor drone; UAV; UAS; VIO; stereo-VIO; vislam; remote sensing

## 1. Introduction

Over the last decade, unoccupied aircraft systems (UAS, drones) have gained much popularity. Although flights under visual line-of-sight control are currently the most common, autonomous flights and beyond visual line-of-sight (BVLOS) operation are enabling a new generation of autonomous applications ranging from the delivery of commercial packages and medical supplies, surveying and mapping to military operations and logistic missions [1]. The availability of reliable position and orientation information is crucial for the autonomous navigation of drones [2]. Drones rely on different types of sensors to obtain odometry information and for navigating through known or unknown environments. The most commonly used positioning approach is to use global navigation satellite system (GNSS) receivers and an inertial measurement unit (IMU). The code-based GNSS positioning provides a few meters accuracy whereas the Real-Time Kinematic (RTK) carrier-based method with differential GNSS (DGNSS) correction improves the positioning accuracy to a centimetric level [3]. However, GNSS positioning is not sufficiently reliable as the accuracy is degraded or the signal may be blocked by the presence of buildings, trees, or due to the

reflection of the signals from walls or other objects [4,5], and might also be subjected to intentional interference [6]. Furthermore, the RTK GNSS positioning requires real-time correction signals from the ground station or CORS/VRS services (Continuously Operating Reference Station/Virtual Reference Station); the interruptions in the connection reduce the accuracy [7]. Complementary positioning systems are thus required in autonomous systems to improve the reliability of positioning to enable secure and reliable operations in various environments [8].

Visual odometry (VO) is an odometry technique used in various domains, such as robotics, automotives, and wearable computing. It is the process of estimating the ego-motion of a system using the input from a single or multiple cameras attached to it [9]. Earlier works have explored the idea of using images from a monocular camera or stereo cameras mounted on a drone to calculate ego-motion using VO [10–15]. Other studies [16–18] have mapped the region along with the path and developed visual simultaneous localisation and mapping (SLAM) algorithms. SLAM is the process of simultaneously building a map of the environment and localising the system in that map [19]. Inertial data can be incorporated with visual data to increase the accuracy and robustness, resulting in VIO and visual-inertial SLAM (VISLAM) solutions [20–23]. Recently, a comprehensive optimization-based estimator for the 15-D state of a drone, a visual-inertial-ranging-lidar (VIRAL), was proposed [24]; data from IMU, ultrawideband (UWB) ranging sensors, and multiple onboard visual-inertial and lidar odometry subsystems were fused in order to eliminate the drift in the solution. A novel VISLAM algorithm performs state estimation even in low illuminated and low textured environments by utilising line features along with the point features, rather than using only the point features in the images [25]. Another study developed a robust VISLAM framework to provide accurate results even in the presence of dynamic and temporary static obstacles, which was not specifically addressed in previous studies [26]. There has also been significant developments in the utilization of deep neural networks to solve the odometry problem [27].

Integrating VO/VSLAM algorithms with outdoor drones enables them to perform autonomous flights without relying on GNSS. Open-source implementations of VO and VSLAM algorithms are already available, and they have been evaluated with the data collected using drones. Previous studies have compared state-of-the-art VIO algorithms with publicly-available drone datasets [28] and implemented autonomous drone navigation using the odometry output from the VINS algorithm [29–31]. In most of the VIO/VISLAM studies, the drone has been flown inside building or at low altitude [24,32–34]. Only a few studies have implemented these algorithms for drones flying at high altitudes, such as altitudes of 40–100 m. A stereo visual pose estimation adapting the traditional stereo-VO paradigm was presented in [12]. Their results for a 2 m wingspan fixed-wing unoccupied aerial vehicle flying at 30–120 m altitude over a 6.5 km trajectory indicated that stereo-VO outperformed the monocular VO; however, accuracy was still relatively low. Modifications were proposed to the state-of-the-art ORB-SLAM2 algorithm in [35] to improve its reliability; the initial algorithm did not manage to reconstruct the trajectory with the DJI Phantom 4 based video.

VO and VSLAM algorithms are considered relative positioning solutions. If the starting position is known in the global Earth fixed coordinate system, the relative trajectory could be translated to the global coordinate system as well. The challenge with relative positioning methods is their drifting. Therefore, mainstream solutions for absolute positioning without GNSS complement the VO solutions by matching the drone images to existing maps, orthophotos or satellite images [36]. These methods are not as sensitive to drifts but the low drift performance is advantageous as it improves robustness, e.g., to potential failures in matching the drone images to the reference datasets.

The objective of this study was to investigate stereo-VIO for high flying altitude drone operation. Due to the lack of suitable sensor suites, a new stereo-visual-inertial sensor system capable of providing useful data at high altitudes was developed. A state-of-the-art VIO algorithm, the VINS-Fusion, was used for testing. In the preliminary study, we

evaluated several algorithms, the ORBSLAM-3, FLVIS and VINS-Fusion [37]; the VINS-Fusion outperformed other algorithms, thus further studies were focused on it. The performance of the system in different configurations, including stereo-VO, mono-VIO and stereo-VIO, at flight altitudes ranging from 40 m to 100 m with respect to ground truth data, was assessed in this study. Since similar data are not publicly available to develop and/or test localisation systems for high flying altitude drone flights, we publish the data collected for this study, accompanied with sensor calibration details and the ground truth information.

## 2. VINS-Fusion—A Stereo-Visual-Inertial Odometry Algorithm

VINS-Fusion is an extension of VINS-Mono, which is a tightly coupled, non-linear optimisation-based state estimation method based on fusing data from a minimum set of sensors [20]. In VINS-Mono, IMU measurements are pre-integrated and fused with features extracted from monocular camera images to estimate the 6 degree-of-freedom (DOF) position and orientation of the robot or the sensor suite. It also enables relocalisation, has a robust initialisation procedure, 4-DOF pose graph optimisation, map merging and map reuse features. In VINS-Fusion, apart from a monocular camera and an IMU, additional sensors, such as cameras and LIDARs, can be integrated seamlessly to improve estimation results, given sensor measurements are modelled as general residual factors, and these residual factors are added to the cost function of the optimiser. Both VINS-Fusion and VINS-Mono share a similar pose estimation pipeline, but VINS-Fusion is designed as a general framework which supports multi-sensor combination [38]. The main stages of the VINS-Fusion algorithm, the data collection, initialization, estimation, relocalisation, and global pose estimation are described in the following.

### 2.1. Data Collection and Preprocessing

Each sensor is considered as a factor that provides a measurement in VINS-Fusion. In this study, the types of sensors were limited to visual and inertial. An overview of the framework is shown in Figure 1. In the case of a stereo camera and IMU suite, the inertial data are collected at a higher frequency than the image capture rate. This ensures multiple inertial measurements between two consecutive images. Shi–Tomasi corners are extracted from the input stereo images and they are matched in the left and right image. As new images are received, the detected features are tracked using the Kanade–Lucas–Tomasi (KLT) sparse optical flow algorithm [39]. Sufficient corner features are detected to maintain a minimum number of features in each frame while enforcing uniform feature distribution over the frame. Keyframes are selected based on the number of tracked features and the average parallax of tracked features in the input images.

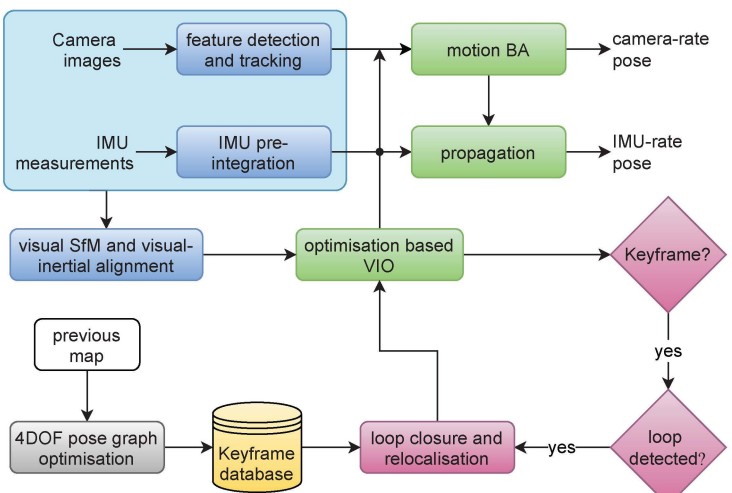

**Figure 1.** Pipeline of VINS-Fusion. Adapted from [20].

The IMU data collected between two consecutive image frames were pre-integrated to calculate relative velocity and rotation of the IMU. Pre-integration of IMU reduces the need of repropagating the states during optimisation, which is a computationally expensive process. As explained in [20], the relative velocity and rotation can be calculated using the IMU measurements and the biases of the accelerometer and gyroscope. The biases of the accelerometer and gyroscope are modelled as random walk which are estimated continuously. Repropagation of IMU values is performed only if the bias estimation changes the bias values significantly. Since IMU propagation is not needed repeatedly, the pre-integration method reduces the computational resource usage in optimisation-based algorithms.

### 2.2. Initialisation

The highly nonlinear tightly coupled VIO have to be initialised accurately at the beginning by loosely aligning the pre-integrated IMU values with the vision-only structure [20]. To accomplish this alignment, rotation and translation between the latest frame and one of the previous frames in the sliding window is computed using the five-point algorithm. There should be sufficient parallax and stable feature tracking between these two frames. An arbitrary scale is set and 3D points are triangulated. Also using perspective-n-point (PnP) algorithm, poses of all the other frames in the window are determined. Finally a full global bundle adjustment is applied to minimize the total reprojection errors of all observed features. This vision-only structure-from-motion (SfM) generates a pose graph where the only parameter is the arbitrarily set scale.

The IMU initialisation results in calculating the gyroscope bias, initialising velocity, gravity vector and metric scale, and gravity refinement. Gyroscope is calibrated using the relative rotation between two consecutive frames calculated by visual SfM and the results of IMU pre-integration. This new gyroscope bias is used to repropagate all the IMU pre-integration terms. The velocity, gravity vector and the metric scale are initialised using the information obtained from the SfM and the new IMU pre-integrated terms. The gravity vector is refined by constraining its magnitude to the known value. Finally, rotation between the world frame and the camera frame are computed by rotating the gravity vector to the $z$ axis, and all the variables from the reference frame to the world frame.

### 2.3. Estimation

A sliding window based state optimisation is performed in VINS-Fusion where the nonlinear least square problem of minimising the cost function is solved using Newton-Gaussian or Levenberg-Marquardt approaches. Ceres solver [40], an open source C++ library for modeling and solving large optimisation problems, is used in the implementation. The state vector includes the position and orientation of the body in the world frame, depth of each feature observed in the first frame, and the IMU measurements corresponding to each image in the sliding window which includes the position, velocity, rotation and the IMU biases. As the number of states increases with time, a marginalisation process is incorporated to reduce the computational complexity. This process removes previous measurements as new measurements are added to the optimisation sliding window, but retains useful information from the measurements being removed. In the implementation of VINS-Fusion, ten key frames are maintained in the sliding window. When a new key frame is added, the visual and inertial factors associated with the first frame are marginalised out.

### 2.4. Relocalisation

The sliding window with marginalisation method of estimation introduces drift that accumulates over time. A tightly coupled relocalisation technique is incorporated to reduce this drift. The first step for relocalisation is the loop detection. The state-of-the-art library for converting images into a bag-of-words representation, DBoW2 [41], is used for loop detection. The features are detected and described using binary robust independent elementary features (BRIEF) descriptors [42]. Upon detecting geometrical and temporal consistency between the descriptors, DBoW2 returns the loop-closure candidates. Relocali-

sation process aligns the sliding window to previous poses where poses of all loop-closure frames are constant. After this, all the IMU measurements, visual measurements and the feature correspondences in the sliding window are optimised jointly which reduces or eliminates the drift.

### 2.5. Global Pose Estimation

A global pose-graph optimisation is performed in addition to the pose-graph optimisation performed in the sliding window to ensure that the set of previous poses are registered in a globally consistent manner. The roll and pitch are fully observable in visual-inertial systems. They are absolute states in the world frame while the position and the yaw values are relative estimates in the reference frame. This results in drift accumulation only in $x$, $y$, $z$, and the yaw angle. So, the pose-graph optimisation is performed after fixing the roll and pitch values, which is a 4-DOF optimisation.

VINS-Fusion supports map merge, save, and load features. For every keyframe, a node is added to the graph with its pose information. Edges between the nodes are defined by either the relative transformation between two frames or if the frames are relocated with loop closure. Along with the node, feature descriptors of corresponding frames are also saved. This graph can be saved and later loaded, which can be connected with a new graph whenever another frame is relocated with one of the nodes in the loaded graph.

## 3. Platform Development

### 3.1. Hardware

VINS-Fusion supports various sensor configurations including monocular vision with inertial data, stereo vision data, and stereo vision with inertial data. In order to leverage the maximum potential of VINS-Fusion, this work used a stereo camera and IMU setup. This section describes the developed sensor suite, hardware and software configurations, calibration of sensors and integration of this system on a commercially available drone.

Off-the-shelf sensor suites for visual-inertial systems are readily available on the market. Intel RealSense T265 packs two fisheye cameras, an IMU and a video processing unit, which provides raw sensor data along with accurate tracking results [43]. Nerian Karmin3 stereo camera series offers stereo cameras with different baselines, which can be paired up with their SceneScan system to extract data from the cameras synchronously [44,45]. Even though these systems allow quick testing by eliminating the need of developing a sensor suite, those were not suitable for this study. The T265 has a very short baseline of 10 cm while the Karmin3 series offers a maximum baseline of 25 cm (Nerian also provides custom baseline stereo setup). A shorter stereo baseline results in larger depth error, and the depth uncertainty increases as the height increases. To minimise these issues, a custom sensor suite with an adjustable baseline and fully controlled components was developed to collect the visual and inertial data.

Along with cameras and IMU, a suitable computing device of a small form factor was chosen. Following factors were considered while developing the system. For an easy integration of the sensor suite with the computer, the computer should be compatible with Linux to run robot operating system (ROS) [46] in it, and the availability of ROS drivers for sensors is desired. The weight of the system should be less than the payload capacity. The camera should have a global shutter sensor, and availability of manual triggering, exposure control and binning are desired. The IMU should be able to output the data at a frequency of at least 100 Hz and it should have trigger control options for synchronisation. Availability of optional RTK corrected GNSS data are preferred as it can be used for generating ground truth data, and also as the major positioning technique while the GNSS satellites are available.

The sensor system comprises two Basler acA2440-75uc [47] cameras and one Xsens MTi-680G RTK GNSS/INS [48] IMU which are mounted on an aluminium bar. This sensor system is connected to GIGABYTE GB-BSi5H-6200-B2-IW (rev. 1.0) mini computer [49] universal serial bus (USB) 3.0 interface. This system is shown in Figure 2.

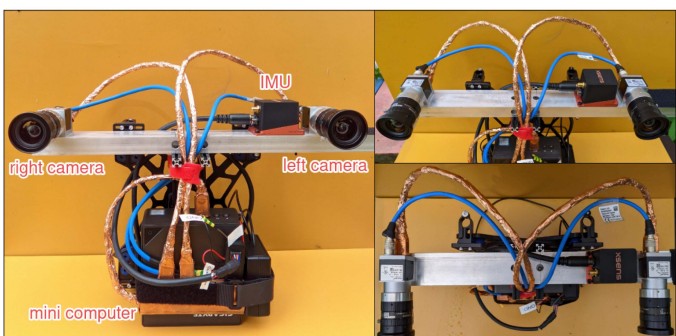

**Figure 2.** Stereo camera and IMU setup connected to the Intel NUC mini computer.

The Basler cameras output colour images of 2448 × 2048 resolution at maximum 75 frames-per-second (FPS). It has a 2/3″ global shutter sensor of size 8.4 mm × 7.1 mm, and a physical pixel size of 3.45 μm × 3.45 μm. Basler provides a ROS package along with camera application programming interfaces (API) for interfacing their cameras easily with the ROS environment. A Fujinon HF6XA-5M 1:1.9/6mm lens was attached to each camera [50]. It was a 6.23 mm fixed focus length lens with maximum resolution of 3.45 μm pixel pitch. The cameras were configured to output monochrome images since the algorithms in this study use monochrome images. The frame rate of the camera was also reduced to the range 15–25 FPS in order to obtain a realtime performance.

The Xsens MTi-680G RTK IMU is capable of streaming out inertial data at a frequency of 400 Hz with an accuracy of 0.25° for roll and pitch values, 0.5° for yaw value [48]. The sensor system was synchronised without using any signals from an external triggering circuit. The Xsens module generated trigger signals based on its internal 400 Hz SDI sampling clock. This signal, with desired skip factor (to reduce the output frequency), triggered the right camera through its input GPIO pin. The right camera sent trigger signals to the left camera when it captured an image. Since there were no external triggers present, the synchronisation was not exact. On average, a temporal difference of 200 μs was present between the corresponding stereo images.

The universal serial bus (USB) 3.0 devices caused interference to wireless devices with lower radio frequencies [51]. The cameras transferred data through USB 3.0 cables which interfered with the GNSS signals by reducing the signal-to-noise ratio of GNSS signals. In order to reduce or fully eliminate this problem, camera cables and the USB 3.0 hub were shielded using copper foil tape.

Cameras were fixed on an aluminium channel where the position of one of the cameras could be adjusted to predefined points to have multiple baselines. In this study, cameras were positioned in the same direction at a distance of 30 cm to form the stereo pair. The IMU was placed in between the cameras. The transformations between sensors were obtained after the calibration. The mini computer and the aluminium channel on which the sensors were mounted were attached to a frame which could be easily attached to the drone. The frame was attached without using a gimbal as shown in Figure 3.

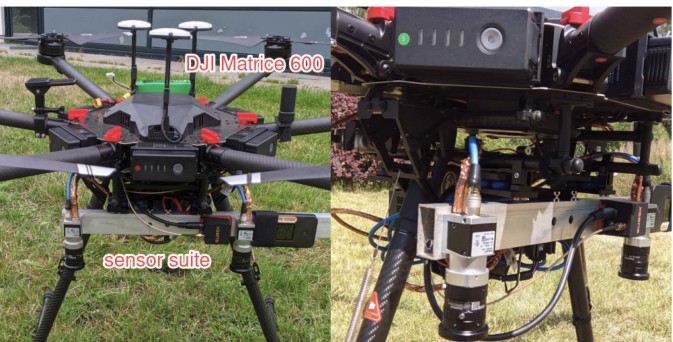

**Figure 3.** Sensor suite and the mini computer mounted on the drone.

### 3.2. Software

Data from both cameras and the IMU were processed in VO and VIO modes. ROS version of VINS-Fusion was available, and sensor drivers were compatible with ROS. Basler provides a ROS package to interface with their cameras. The `pylon_camera` package is a ROS wrapper for their C++ application programming interfaces (API) [52] which can be used to retrieve images from and to configure various parameters of the camera. Images from the camera were of size $2448 \times 2048$ px (5 Mpix). To have a real time performance, images were subsampled to reduce their size to $612 \times 512$ px (0.3 Mpix). The VINS-Fusion algorithm expects both images to have the same timestamp. The short temporal difference of 200 µs between the corresponding stereo image pairs was neglected, and timestamps of images were set to the same value before they are processed further.

IMU data were acquired from the sensor using the `xsens_mti_ros_node` ROS package provided by Xsens [53]. RTK corrections provided by FinnRef, a Finnish network of reference stations [54], were streamed to the mini computer using NTRIP whenever available. The RTK corrected GNSS data could be used as ground truth information if the corrections were streamed while collecting the data.

### 3.3. Calibration

Stereo cameras and the IMU were calibrated to obtain their intrinsic and extrinsic parameters. A multisensor calibration toolbox, `kalibr` [55], along with `imu_utils` [56], was used to calibrate the sensor suite. `imu_utils` produced IMU noise model parameter values. Images were subsampled to 0.3 Mpix before using them for calibration. Stereo camera calibration was performed to calculate intrinsic and extrinsic parameters for each camera using the Kalibr toolbox [57]. Finally, using the above two results, a stereo camera and IMU joint calibration was performed in Kalibr [58]. All the sensor parameters were published along with the sensor data.

The IMU measurement model used in Kalibr for stereo camera and IMU joint calibration contained two types of errors: additive noise and sensor bias. The angular rate measurement, $\widetilde{\omega}$, was modelled as

$$\widetilde{\omega}(t) = \omega(t) + b(t) + n(t). \tag{1}$$

Here, $\omega$ was the actual angular rate. This measurement model was for a single axis of gyroscope. The fluctuations in the sensor signal were modelled with a zero-mean continuous-time white Gaussian noise $n(t)$ and the variations in sensor bias were modelled with a random walk $b(t)$. Both of these parameters were calculated for the accelerometer and the gyroscope of IMU. These noise model parameters were obtained by analysing the Allan standard deviation of the IMU data [59]. This was performed by the `imu_utils` tool with the IMU data collected while the IMU was kept stationary for a duration of two hours.

For the stereo camera calibration, stereo image pairs were recorded at a low frame rate of 4 FPS as a Bag file. An Aprilgrid board [60] was used as the calibration target. The camera was kept stationary and the board was moved while recording the data. Patterns on the Aprigrid targets could be detected individually, which simplified the data collection as the calibration could be performed even if the target was only partially visible in the image.

A joint IMU-stereo camera calibration was performed to calculate the extrinsic parameters, namely the rotation and translation of cameras with respect to the IMU frame. In this case, the calibration target, Apriltag, was kept stationary and the sensors were moved while collecting the data. Image pairs were collected at a 20 FPS rate and the IMU data were collected at 100 Hz rate. Sensors were moved in such a way that all rotations and accelerations axes of the IMU were excited. This recorded data, along with the results of individual IMU and stereo camera calibrations, were processed together in Kalibr to calculate the transformation between the IMU and the cameras.

## 4. Experiments

The drone was flown near the FGI office building in Masala, Finland, in predefined paths marked by waypoints to collect the data. These waypoints were defined in UgCS drone mission planning and flight control software [61] which generated the mission plan. Flight altitude and flight speed were modified and uploaded to the drone before each flight using this software. The mapping area comprised a building, parking area, road and forest. The area above which the drone was flown is shown in Figure 4. This image was reconstructed using the images from the 60 m, 2 m/s dataset in Agisoft Metashape software [62]. Other datasets were also collected by flying the drone in similar paths. Various steps of the experiments are described in Algorithm 1 and they are detailed in below subsections.

---

**Algorithm 1** Experiment.

---

1: Define and upload the flight mission containing waypoints and flight speeds to the drone.
2: Start the mission.
3: Record sensor data till the mission ends.
4: Collect ground control point (GCP) locations for reference.
5: Run VINS-Fusion in different sensor configurations with the collected sensor data and store the estimated drone trajectory.
6: Generate the ground truth data with Agisoft Metashape software using the stereo images and the GCP data.
7: Compare the estimated trajectory with the ground truth by calculating the error metrics.

---

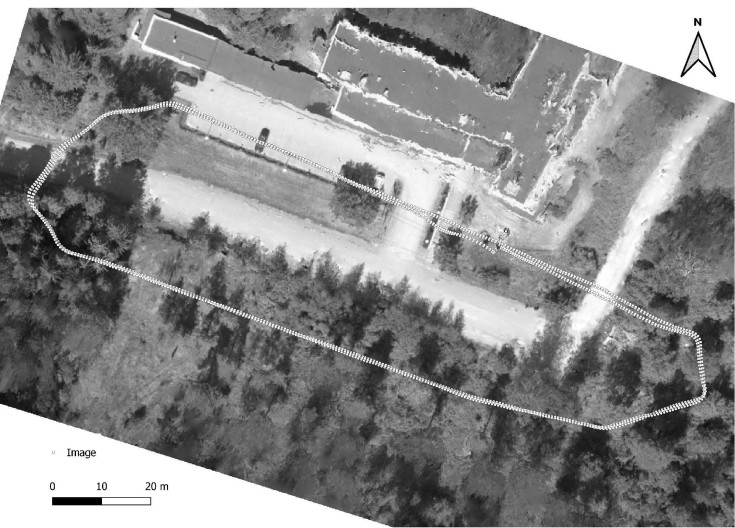

**Figure 4.** The area above which the datasets were collected. This image was reconstructed using the images from the 60 m, 2 m/s dataset in Agisoft Metashape software. The flight path is marked in the image.

### 4.1. Drone Data Collection

The frame rate of both cameras and the IMU output frequency were kept fixed for each dataset at 16 FPS and 100 Hz, respectively. The exposure time of the cameras was adjusted based on the lighting conditions before each flight. These values, along with flight parameters and external conditions, are given in Table 1. Each dataset in this table was collected on different days, due to which external factors such as lighting and wind conditions were not the same during each flight. Sample images from each dataset are shown in Figure 5. Dataset 2 was over exposed while datasets 3 and 4 were underexposed. Cameras and IMU output the data to the Intel NUC minicomputer over the ROS network. Recording of sensor data started at an arbitrary time during takeoff and ended during landing, and varied for each dataset. These sensor messages were stored as bag files in the mini computer for further processing.

**Table 1.** Recorded datasets.

| Altitude (m) | 40 | | | 60 | | | 80 | | | 100 | | | Exposure Time (μs) | External Conditions [1] |
|---|---|---|---|---|---|---|---|---|---|---|---|---|---|---|
| Speed (m/s) | 2 | 3 | 4 | 2 | 3 | 4 | 2 | 3 | 4 | 2 | 3 | 4 | | |
| Dataset 1 | √ | | | √ | √ | | | | | | | | 500 | gentle breeze |
| Dataset 2 | | √ | | | | | | √ | | | √ | | 1000 | moderate breeze |
| Dataset 3 | | | | √ | | | | √ | | | | √ | 1000 | cloudy, moderate breeze |
| Dataset 4 | | | √ | | | | √ | | | √ | | | 1200 | cloudy, strong breeze |

[1] Wind speed data collected from Finnish Meteorological Institute's website, and mentioned in Beaufort scale.

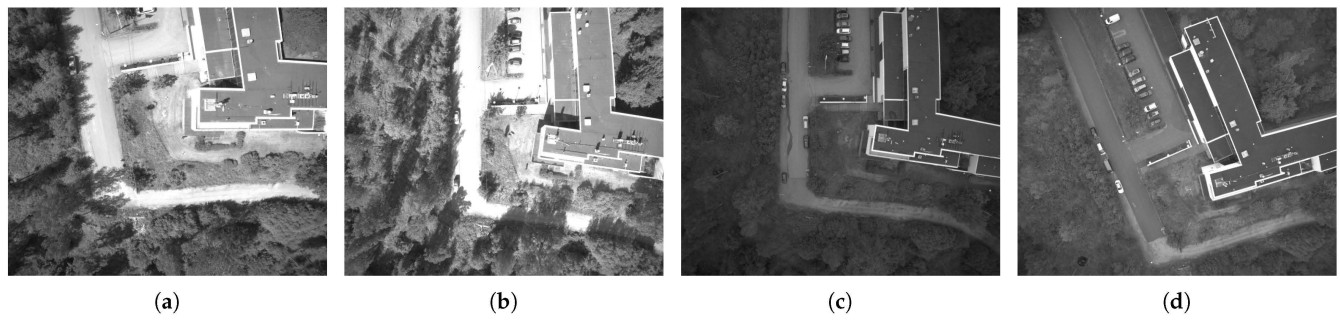

(**a**)　　　　　　　　(**b**)　　　　　　　　(**c**)　　　　　　　　(**d**)

**Figure 5.** Sample images from collected datasets. Images in each dataset has different illumination due to different lighting conditions during data collection (**a**) Dataset 1—normal exposure. (**b**) Dataset 2—overexposed. (**c**) Dataset 3—underexposed. (**d**) Dataset 4—underexposed.

### 4.2. Data Processing

The VINS-Fusion algorithm was evaluated in offline processing mode in which the collected sensor data were processed later to estimate the trajectory. Open sourced ROS package implementation of VINS-Fusion was installed and was configured with sensor calibration details. Additional parameters related to feature tracking and optimisation were configured as given in Table 2.

**Table 2.** VINS-Fusion parameters.

| Feature Tracking Parameters | Value | Optimisation Parameters | Value |
|---|---|---|---|
| Max number of tracked features | 150 | Max solver iteration time | 0.04 ms |
| Min distance between two features | 30 px | Max solver iterations | 8 |
| RANSAC threshold | 1 px | Keyframe parallax threshold | 10 px |

The collected data were given as input to the algorithm. Three different sensor configurations—monocular camera and IMU (mono-VIO), stereo camera (stereo-VO), and stereo camera and IMU (stereo-VIO)—were tested while estimating the trajectory. The estimated poses were in local coordinates where the origin was the starting point of the flight.

### 4.3. Reference Trajectory

The reference trajectory was obtained by post-processing the recorded images along with the ground control points (GCP) in Agisoft Metashape software [62]. Previous studies

have shown that estimation results from Metashape are highly accurate, given that the image block has sufficient geometric structure and accurately measured GCPs are available for processing [63]. The GCPs were targeted with the help of markers placed in the mapping area. Positions of these markers were recorded using a HiPer HR GNSS receiver and an FC-5000 Field Controller by Topcon Positioning Systems [64,65]. In this work, photogrammetric bundle block adjustment was performed on the stereo images in Metashape to obtain the position and orientation of cameras during the image acquisition. Since images were recorded at 16 FPS rate, adjacent images have high overlap, especially when the drone had flown at a low speed of 2 m/s. Therefore, the images were subsampled, and this small sample along with the ground control points were processed. Output of this process includes the pose of the camera for each image. Poses were generated in both local and geographic coordinate systems.

### 4.4. Error Metrics

The quality of the estimated trajectories was evaluated and analysed to understand and benchmark the performance of the algorithm at different flight conditions and with different system configurations. Studies have proposed absolute trajectory error (ATE) and relative pose error as error metrics to evaluate VO/SLAM algorithms [66]. More informative results were obtained by calculating relative error (RE) for each sub-trajectory of the estimation [67]. A quantitative performance evaluation was carried out by computing ATE and RE with respect to the ground truth [68]. It follows a two step approach in which the estimated trajectory is aligned with the ground truth initially, and the error metrics are calculated based on the aligned trajectory.

The trajectory alignment was performed based on the Umeyama method [69], which aligned both the ground truth and the estimated trajectories based on multiple estimated poses of the trajectory. The alignment process calculated a rotation and translation (and scale in case of monocular VO) which was applied to the estimated trajectory to align it with the ground truth.

The ATE of position and rotation gave a single number metric corresponding to the estimation. For a single state, the error from the ground truth was parameterised as

$$\Delta \mathbf{x} = \{\Delta \mathbf{R}, \Delta \mathbf{p}, \Delta \mathbf{v}\}, \tag{2}$$

where $\Delta \mathbf{R}$, $\Delta \mathbf{p}$ and $\Delta \mathbf{v}$ correspond to the rotation, position and velocity errors respectively. For the entire trajectory, the root mean square error (RMSE) value was calculated to obtain a single metric value.

$$\mathrm{ATE_{rot}} = \left( \frac{1}{N} \sum_{i=0}^{N-1} \| \angle (\Delta \mathbf{R}_i) \|^2 \right)^{\frac{1}{2}} \tag{3}$$

$$\mathrm{ATE_{pos}} = \left( \frac{1}{N} \sum_{i=0}^{N-1} \| (\Delta \mathbf{p}_i) \|^2 \right)^{\frac{1}{2}}. \tag{4}$$

If there was a rotation error at the beginning of the trajectory, the final $\mathrm{ATE_{rot}}$ value would be higher compared to that corresponding to the case where the rotation error occurred at the end of the trajectory [68]. Since ATE was sensitive to the time of error occurrence, RE, which provided more information about relative changes, was also calculated along with ATE in this work.

The quality of estimations can also be evaluated by measuring relative relations between the states at different times since there exists no global reference for VO/VIO systems [68]. RE was computed by splitting the trajectory into small sub-trajectories, aligning each of them separately, and calculating the RMSE of the errors between the ground truth and the aligned sub-trajectory, similar to the computation of ATE. Sub-trajectories were defined using a set of pairs of states, $d_k = \{\hat{\mathbf{x}}_s, \hat{\mathbf{x}}_e\}, e > s$. These pairs were

selected based on some criteria (e.g., distance along the trajectory). For each sub-trajectory, the errors were parameterised as

$$\delta \mathbf{d}_k = \{\delta \phi_k, \delta \mathbf{p}_k, \delta \mathbf{v}_k\}. \tag{5}$$

Rotation, position and velocity errors in $\delta \mathbf{d}_k$ were calculated similar to ATE, and the error calculation was performed for all the sub-trajectories. Unlike ATE, the relative error provided a number as a metric for each sub-trajectory as given below.

$$\text{RE}_{\text{rot}} = \{\delta \phi_k\} \tag{6}$$

$$\text{RE}_{\text{pos}} = \{\delta \mathbf{p}_k\}. \tag{7}$$

Estimated trajectories were evaluated based on error metrics $\text{ATE}_{\text{pos}}$ and $\text{RE}_{\text{rot}}$. These metrics were calculated using an open-source tool, `rpg_trajectory_evaluation` [70], which is based on [68]. Outputs of this tool included ATE and RE values along with their plots. To get a single value for the relative error, a combined RMSE value was calculated from each individual RE.

## 5. Results and Discussion

### 5.1. Experiment Results

The collected sensor data were processed using the VINS-Fusion algorithm in offline mode to obtain the trajectory. The estimated trajectories of three different sensor configurations were compared with the ground truth. Corresponding error metrics are given in Table 3. Values for the mono-VIO case of 40 m and 2 m/s case were not available as the estimated trajectory did not resemble the reference trajectory.

**Table 3.** ATE of position and RE of yaw for the data collected at different flight parameters.

| Speed | Estimation | 40 m ATE | RE | 60 m ATE | RE | 80 m ATE | RE | 100 m ATE | RE |
|---|---|---|---|---|---|---|---|---|---|
| 2 m/s | stereo-VO | 9.944 | 2.291 | 15.668 | 3.651 | 62.415 | 14.276 | 101.584 | 9.006 |
| | mono-VIO | – | – | 55.814 | 10.394 | 12.093 | 1.634 | 18.409 | 2.582 |
| | stereo-VIO | 21.689 | 3.627 | 10.563 | 0.949 | 4.362 | 1.414 | 12.861 | 1.497 |
| 3 m/s | stereo-VO | 16.932 | 6.436 | 14.092 | 3.127 | 56.992 | 8.953 | 123.113 | 9.393 |
| | mono-VIO | 9.462 | 3.575 | 16.547 | 2.182 | 6.741 | 4.483 | 20.302 | 4.01 |
| | stereo-VIO | 2.554 | 2.638 | 4.744 | **0.862** | 7.397 | 4.034 | 12.459 | 4.069 |
| 4 m/s | stereo-VO | 16.183 | 2.454 | 41.240 | 5.731 | 71.668 | 3.651 | 113.007 | 3.388 |
| | mono-VIO | 9.957 | 2.323 | 15.495 | 4.883 | 20.256 | 2.27 | 17.909 | 6.164 |
| | stereo-VIO | 4.636 | 1.312 | **2.186** | 3.214 | 9.245 | 1.819 | 9.544 | 4.435 |

The accuracy of estimation using stereo-VO for the same flight speed degraded as the flying altitude increased. Plots of estimated trajectories corresponding to 3 m/s flight speed are shown in Figure 6a. A similar trend was seen in other cases as well, where the flight speeds were 2 m/s and 4 m/s. Increased drift in position estimations with increasing flight altitude was due to the decreasing baseline-to-depth ratio of the stereo setup, and it was the expected behaviour. The error built up quickly due to a poorly triangulated structure, and the scale was observed weakly at high altitudes due to the constrained stereo baseline [71]. The yaw rotation errors in stereo-VO estimation were higher than 2° for all the cases. These error values were not correlated to flying altitudes or speeds.

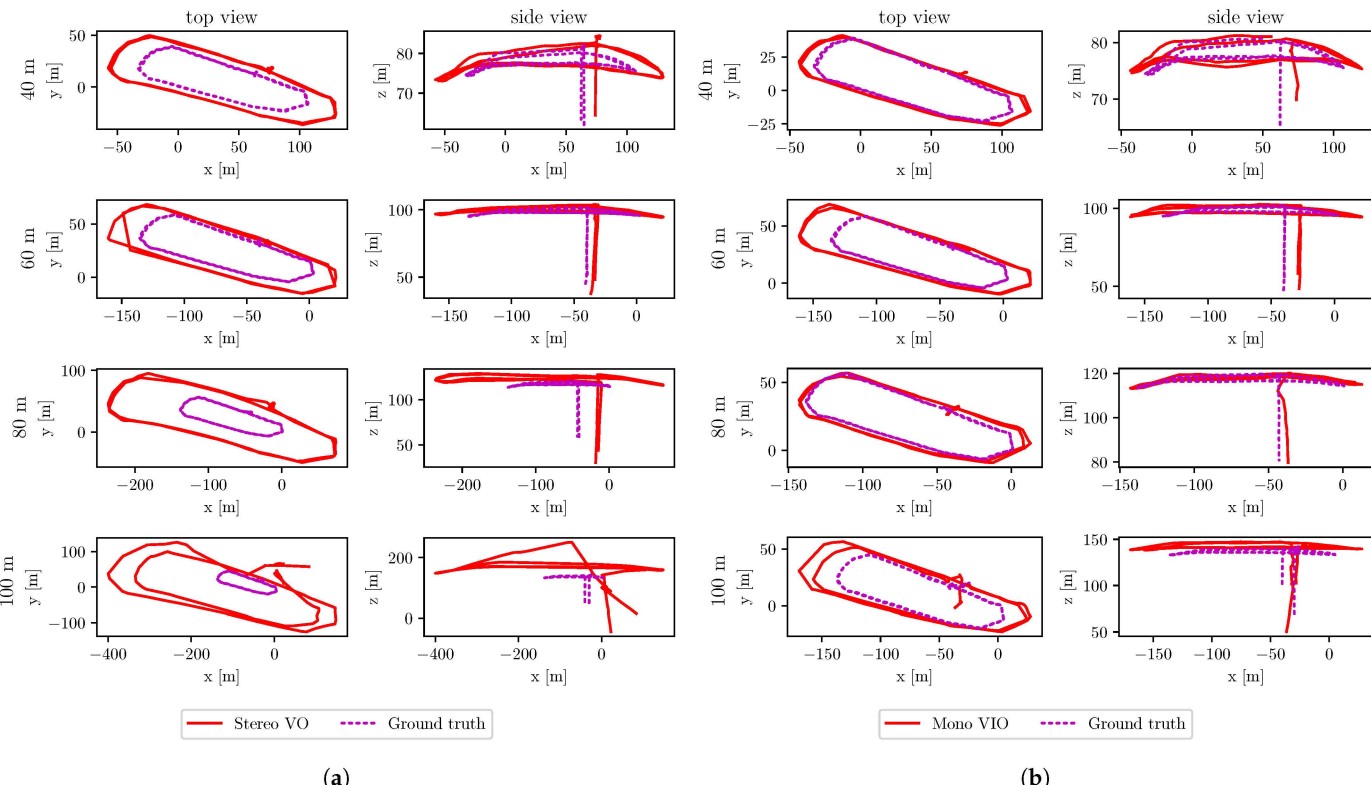

**Figure 6.** Trajectory estimation by VINS-Fusion plotted along with the ground truth for different altitudes at a speed of 3 m/s. (**a**) Stereo visual odometry estimation. (**b**) Mono visual-inertial odometry estimation.

Fusing data from an IMU with the visual inputs improved the estimation accuracy, especially when the conditions for visual estimation were not good. The mono-VIO estimation had a lower position error compared to the stereo-VO estimation, except for the data in dataset 1; which had better illumination compared to other datasets where images were underexposed or overexposed as shown in Figure 5. This resulted in getting better results for vision only estimation, but only when images had good illumination. For other datasets, results were better when data from IMU and a camera were used; both rotation and position errors were lower for the mono-VIO estimation than stereo-VO estimation as shown in Table 3. Plots of estimated trajectories corresponding to 3 m/s flight speed are shown in Figure 6b. Scale errors of estimations at flying altitude 60 m and speed 4 m/s are plotted in Figure 7. A reduced drift in comparison to stereo-VO is seen here as the errors decreased with the availability of data from multiple sensors.

With stereo-VIO estimation, position error values were reduced drastically. The lowest error in position estimate was 2.186 m when the drone flew at 60 m height with 4 m/s speed, and the lowest rotation error was 0.862° when the drone flew at 60 m height with 3 m/s speed. The stereo-VIO produced significantly more accurate position and orientation estimates than mono-VIO and stereo-VO. The trend with the increasing flight height as seen in stereo-VO was not seen for monocular or stereo inertial estimations. In the case of stereo-VIO, at speed 3 m/s, the position error increased as the flying altitude increased, but the rate was only a fraction of that in the stereo-VO case. For other speeds, such a relation was not seen. The ideal flight speed and flight height were 3 m/s and higher, and 40–60 m, respectively. For these cases, RMSEs were 2–5 m for position and 0.9–3.2° for yaw rotation. It can be concluded that the integration of IMU into the trajectory estimation stabilised the trajectory error to an acceptable level. The stereo-VIO estimation plots corresponding to 4 m/s speed are shown in Figure 8.

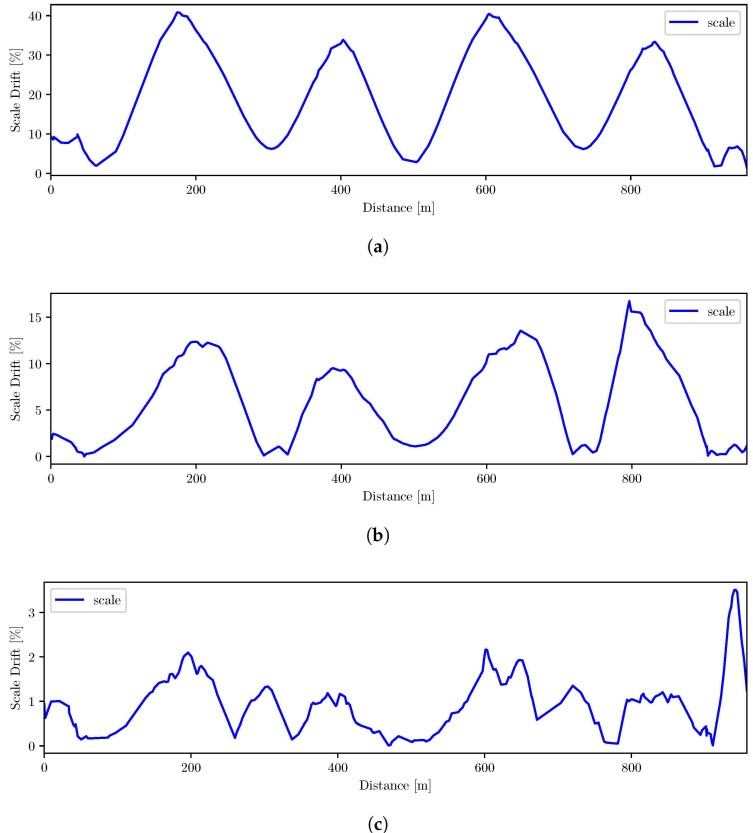

**Figure 7.** Scale errors of estimations at flying altitude 60 m and speed 4 m/s. (**a**) stereo-VO. (**b**) mono-VIO. (**c**) stereo-VIO.

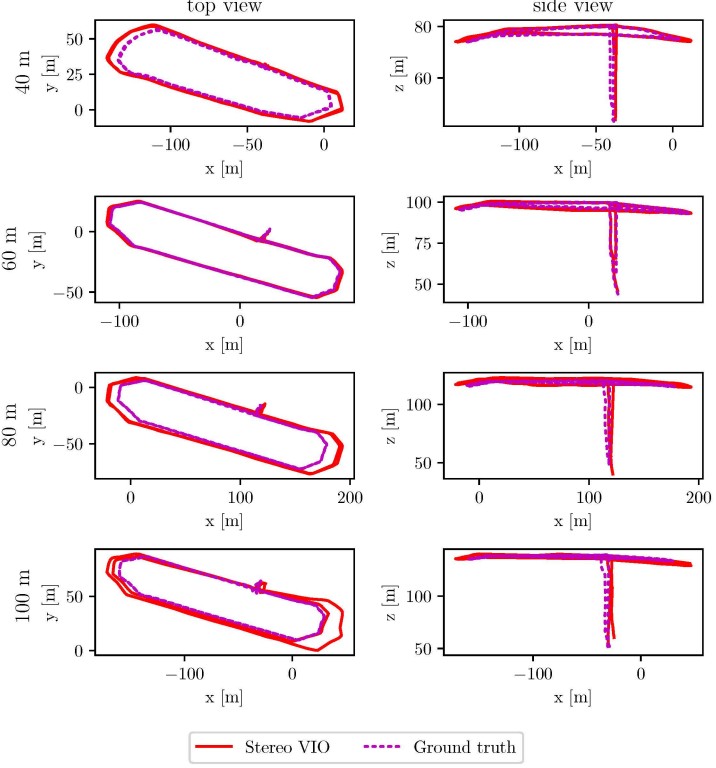

**Figure 8.** Stereo visual inertial odometry estimation by VINS-Fusion plotted along with the ground truth for different altitudes at a speed of 4 m/s.

Illumination and the scene content of the images influenced the estimation quality. The 60 m altitude data were recorded in two days as mentioned in Table 1, where images of 2 m/s and 3 m/s datasets (dataset 1) had good illumination whereas the 4 m/s dataset (dataset 3) had low illumination because of changes in lighting conditions and exposure timings. Even though the position error decreased with increased speed at this altitude, the rotation error increased when the images had low illumination regardless of the flight speed.

Relative yaw error was almost the same for all datasets collected on the same day. For dataset 4, rotation errors for all three sets were around 1.4° while that of dataset 2 was around 4°. For dataset 1, the error for the 40 m, 2 m/s case was larger than that for the other cases. Images in this set had a uniform pattern when the drone flew over the trees at a low altitude. This uniform pattern caused the algorithm to estimate poses less accurately. Absolute rotation errors for the flights at 40 m and 60 m altitude with 2 m/s speed are shown in Figure 9. For the 40 m case, the absolute yaw error remained constant up to 500 m flying distance, keeping the relative error almost zero, but the error increased when the drone flew over trees. Additionally, as the flying altitude increased in the 60 m case, the camera view included distinct objects, which resulted in better feature tracking and pose estimation.

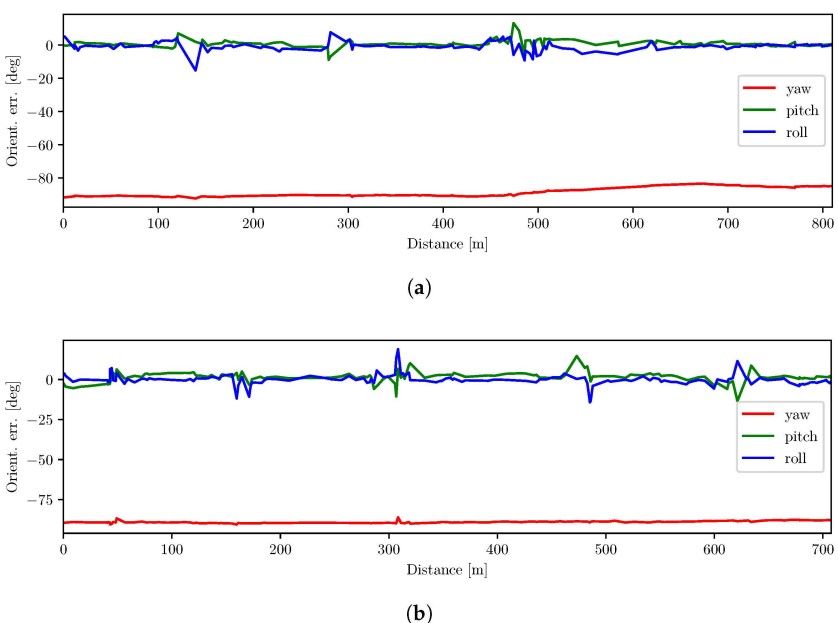

(**a**)

(**b**)

**Figure 9.** Absolute rotation errors at flying speed 2 m/s. (**a**) 40 m flying altitude (**b**) 60 m flying altitude.

IMU measurements had a direct impact on the estimation during the initialisation process as mentioned in Section 2.2. Pre-integrated IMU values were used to calculate the actual scale for triangulation. The velocity and gravity vector were also initialised during this process. Since the initialisation process used the data collected during take-off, jerks or abrupt movements during this time affected the IMU measurements. This might have resulted in estimating less accurate parameters, in turn reducing the overall estimation accuracy. The integration of statistics estimation to the VINS-Fusion would enable the obtaining of theoretical values that could provide further understanding of the system behaviour.

### 5.2. Assessment, Contribution and Future Research

An important contribution of this study was the development of an adjustable sensor system for outdoor stereo-VIO. The results were consistent with respect to theoretical expectations as well as with previous studies carried out at lower flight altitudes and in

indoor conditions. The stereo-VO results were at similar levels to those in [10]. The studies with low altitude flights have shown that the integration of IMU into VO reduces drifts [72]; the empirical results were consistent with this expectation. The decreasing base-to-height ratio is known to deteriorate the performance of stereo-VO that was also observed in the results. The stereo-VIO with a baseline of 30 cm provided significantly better results than the mono-VIO for all flying altitudes of 40–100 m, thus the use of multiple cameras is recommended for the VIO-based positioning. According to the authors' knowledge, this study was the first study to validate these behaviours at a flight altitude range of 40–100 m.

Apart from the sensor configuration, external conditions also affected the estimation accuracy. It was observed that with better illumination conditions, stereo-VO provided better estimations than mono-VIO. Errors were larger when the scene had a uniform texture; flying at a higher altitude to have distinct objects in the frame helped to reduce the estimation error. Fusing IMU data improved positioning performance and improved the robustness of the system to the external conditions. Studying the effects of external factors on IMU, such as jerks or abrupt movements, and thus the effect on estimation quality, would help improve the estimation. External conditions during the data collection were different for each dataset; uniform or similar conditions would have enabled more detailed analysis of impacts of flying heights and speeds. However, experiments with the developed system confirmed reliable estimations with stereo-VIO in a variety of conditions. Consistent performance of the system at various flying heights and speeds indicated that this system would be capable of providing useful pose estimations for GNSS-free operations. Real-time testing would be required to verify this.

This study showed that with the help of suitable stereo-visual-inertial sensor suite and sensor fusion algorithms, a reliable localisation system for high-altitude outdoor drone operations can be developed. It could be used alongside, or replace, existing positioning systems such as GNSS based systems. The system is useful as a redundant positioning system to fill gaps in the trajectory measured by GNSS or other positioning techniques. This study provided new knowledge considering the performance of VIO algorithms at high flying altitudes; this is of great importance because the use of autonomous drones and beyond visual line-of-sight flying is becoming possible due to evolving legislation and will increase enormously in the coming years.

Our future research will continue the development of reliable positioning techniques for autonomous flying. A well-engineered system capable of real-time estimation will enable further studies. New odometry or SLAM algorithms can be implemented on this system—especially since the deep learning based positioning techniques have shown excellent potential [27] in recent years. Additionally, it is relevant to study the integration of additional sensors to further reduce the drifts [24]. Further studies could also evaluate whether the robust positioning with stereo-VIO would outperform commonly used mono-VIO algorithms in positioning systems based on maps or satellite images [36]. For greater flying altitudes, it might be necessary to increase the stereo baseline; this could be evaluated in future studies utilising the adjustable baseline concept available in the system. Future research should also emphasise the system calibration models. We selected models implemented in the well-known tool Kalibr; the models provided consistent results in our work. However, if the models did not correspond to reality, this would result in biases and drifts in the estimations which would reduce the accuracy.

The dataset has been made openly available to facilitate the further research and development of VIO algorithms for high flying altitude drone operation for the scientific community.

## 6. Conclusions

This study investigated visual odometry-based positioning techniques for high-altitude drone operation. An important contribution of this work was the development and presentation of a new stereo-visual-inertial sensor suite based on state-of-the-art hardware components for high flying altitude drone operation as well as empirical studies using this system. In the performance assessment, the system was used to collect sensor data and the

VINS-Fusion algorithm was used for localization and to provide a 6-DOF pose estimation solution at flying heights of 40–100 m. Data from three different sensor configurations—stereo camera, monocular camera and IMU, and stereo camera and IMU—were used to estimate the trajectory of the drone. Stereo-visual-inertial odometry provided best results with the lowest position and rotation errors. The lowest RMSE values of absolute translation and relative yaw error were 2.186 m and 0.862° respectively for a 800 m long trajectory at a flying altitude of 60 m. Apart from the sensor configuration, external conditions such as illumination and object texture also affected the estimation accuracy.

This study provided new knowledge considering the performance of VIO algorithms at high flying altitudes. Research on positioning algorithms for high flying altitude drone flights is of great importance because the autonomous use of drones and beyond visual line-of-sight flying is increasing and this will require redundant positioning solutions that compensate for potential disruptions of GNSS positioning.

The data collected in this study are shared with the research community so that these results can be verified, and the data can be used to develop and test new solutions for high flying altitude drone localisation.

**Author Contributions:** Conceptualization, A.G., E.H. and N.K.; methodology, A.G. and N.K.; software, A.G.; validation, A.G. and N.K.; formal analysis, A.G. and N.K.; investigation, A.G.; resources, A.G., N.K., T.H. and J.S.; data curation, A.G.; writing—original draft preparation, A.G. and E.H.; writing—review and editing, A.G., N.K., T.H., J.S. and E.H.; visualization, A.G.; supervision, T.H., J.S. and E.H.; project administration, E.H.; funding acquisition, E.H. All authors have read and agreed to the published version of the manuscript.

**Funding:** This research was part of part of Academy of Finland projects, Finnish UAV Ecosystem—FUAVE (Decision number 337018) and Autonomous tree health analyzer based on imaging UAV spectrometry—ASPECT (Decision number 327861).

**Data Availability Statement:** The data presented in this study are openly available in Etsin, Fairdata at https://doi.org/10.23729/f4c5d2d0-eddb-40a1-89ce-601c252dab35, accessed on 30 November 2022.

**Conflicts of Interest:** The authors declare no conflict of interest.

## Abbreviations

The following abbreviations are used in this manuscript:

| | |
|---|---|
| API | application programming interface |
| ATE | absolute trajectory error |
| BRIEF | binary robust independent elementary features |
| BVLOS | beyond visual line-of-sight |
| FPS | frames per second |
| GNSS | global navigation satellite system |
| GPIO | general-purpose input/output |
| IMU | inertial measurement unit |
| LIDAR | light detection and ranging |
| PnP | perspective-n-point |
| RE | relative error |
| ROS | robot operating system |
| RTK | real time kinematics |
| SfM | structure from motion |
| SLAM | simultaneous localisation and mapping |
| USB | universal serial bus |
| VIO | visual-inertial odometry |
| VISLAM | visual-inertial SLAM |
| VO | visual odometry |
| VSLAM | visual SLAM |

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
