# Peer review of "Visual-Inertial Odometry Using High Flying Altitude Drone Datasets"

_drones, doi:10.3390/drones7010036_

Round 1

Reviewer 1 Report

Title of the Article: Visual-Inertial Odometry using High Altitude Drone Datasets

In the submitted Article, the implementation and assessment of a redundant positioning system for high altitude drone operation based on visual-inertial odometry was studied.

The author cleanly identifies the gap in the current state-of the art literature here I quote “Only a few studies have implemented the visual-inertial odometry (VIO), the algorithms used in current study for high altitude drones”.

In the literature review, the authors cite the most recent publications (90% of published articles are less than 10 years).

The methodology is well structured as demonstrated by the prototype developed as shown in Fig. 2-3 and the data collection presented in Fig.4-5 and Table. However, the boundary of data collected in Figure 4-5 is not clearly visible. Colored pictures can present data correctly. A short description of data collected in each Figure is required for better understanding.

 The results are well presented and discussed as shown in Fig7-9.

The conclusions are supported by the results. The authors clearly show the gap left in the article and the recommendation for future work is proposed as it reads: “Studying the effects of external factors on IMU, such as jerks or abrupt movements, and thus the effect on estimation quality would help improving the estimation.”

Recommendation: The article is scientifically sound and is recommended for publication after addressing the minor comments.

Reviewer 2 Report

1.- The authors must highlight the contribution of the work, it’s not clear.

2.- In Figures 2-3 it is necessary to remark or underline the components and put legends to a better visualization.

3.- Could be interesting that the authors explain the phenomenon of a random walk in the angular estimation.

4.- What happen with the estimation of angular rate if $\eta$ is not modeled as a white noise?

5.- In Figure 6, the ground truth line must be plotted with a dotted line in order to visualize better the performance. And the same in Figure 8.

6.- It would be good to test the drone at 100 m, since it is appreciated that as it gets higher it presents less errors.

7.- An algorithm of the implementation of each test would be very helpful to understand them better.

Round 2

Reviewer 2 Report

The reviewer’s comments on the last version of this manuscript have been addressed in this revision. It is recommended to be accepted for publication.

Reviewer 3 Report

All my concerns have been addressed.